# Pharmatherapeutic Treatment of Osteoarthrosis—Does the Pill against Already Exist? A Narrative Review

**DOI:** 10.3390/jpm13071087

**Published:** 2023-06-30

**Authors:** Frauke Wilken, Peter Buschner, Christian Benignus, Anna-Maria Behr, Johannes Rieger, Johannes Beckmann

**Affiliations:** 1Department of Orthopedic Surgery and Traumatology, Hospital Barmherzige Brüder Munich, Romanstr. 93, 80639 München, Germany; peter.buschner@barmherzige-muenchen.de (P.B.); anna-maria.behr@barmherzige-muenchen.de (A.-M.B.); johannes.rieger2@barmherzige-muenchen.de (J.R.); johannes.beckmann@barmherzige-muenchen.de (J.B.); 2Department of Traumatology and Orthopedic Surgery, Hospital Ludwigsburg, Posilipostr. 4, 71640 Ludwigsburg, Germany; christian.benignus@rkh-gesundheit.de

**Keywords:** osteoarthritis, cartilage, chondroprotectors, platelet rich plasma, NSAIDs, mesenchymal stem cells

## Abstract

The aim of this narrative review is to summarize the current pharmacotherapeutic treatment options for osteoarthritis (OA). Is therapy still mainly symptomatic or does the pill against arthrosis already exist? Causal and non-causal, as well as future therapeutic approaches, are discussed. Various surgical and non-surgical treatment options are available that can help manage symptoms, slow down progression, and improve quality of life. To date, however, therapy is still mainly symptomatic, often using painkilling and anti-inflammatory drugs until the final stage, which is usually joint replacement. These “symptomatic pills against” have side effects and do not alter the progression of OA, which is caused by an imbalance between degenerative and regenerative processes. Next to resolving mechanical issues, the goal must be to gain a better understanding of the cellular and molecular basis of OA. Recently, there has been a lot of interest in cartilage-regenerative medicine and in the current style of treating rheumatoid arthritis, where drug therapy (“the pill against”) has been established to slow down or even stop the progression of rheumatoid arthritis and has banned the vast majority of former almost regular severe joint destructions. However, the “causal pill against” OA does not exist so far. First, the early detection of osteoarthritis by means of biomarkers and imaging should therefore gain more focus. Second, future therapeutic approaches have to identify innovative therapeutic approaches influencing inflammatory and metabolic processes. Several pharmacologic, genetic, and even epigenetic attempts are promising, but none have clinically improved causal therapy so far, unfortunately.

## 1. Introduction

Osteoarthritis (OA) is a disease with a degenerative and inflammatory component that affects a large proportion of the ageing population [1]. The joints of the lower extremities are particularly affected, especially the hip and knee. Cartilage loss is the most visible change, but all joint structures are affected, some occurring quite a bit earlier than cartilage, such as the synovia or subchondral bone. OA involves a variety of factors, such as mechanical loading, ageing, inflammation, and metabolic changes, and the activation of different signaling pathways and enzymatic processes, ultimately leading to a progressive loss of joint function. Consequently, osteoarthritis is a heterogeneous disease with a common end route but many different starting points. Accordingly, there are diverse treatment modalities. They can either be conservative such as thermal, pharmacological, orthotic or physiotherapeutic, or surgical. The latter can be reconstructive, such as in cartilage stimulation or cartilage transplantation, load allocating, such as in osteotomies, or finally replacing, such as in terms of artificial joint replacement. However, especially in the early stages, the therapeutic approach often remains purely symptomatic. Currently, there is no causal drug for the treatment of OA. The most common treatment form is the use of painkilling and anti-inflammatory drugs until the final stage of treatment, which is usually joint replacement. However, the actual cause of OA is often not being treated. Another recent narrative review summarized the guidelines and symptomatic pharmacotherapeutic treatments [2].

The current review’s aim is to describe future perspectives. In recent years, the pathogenesis of osteoarthritis has become increasingly well understood, and new therapeutic approaches are emerging. There are clear similarities to rheumatoid arthritis, which led to a large number of rapid and severe joint destructions several decades ago. Today, those cases are rare due to new medical approaches, mainly the so-called biologicals [3]. The goal is to implement causal therapy and to abandon singular symptomatic therapy in OA as well. By influencing the underlying signaling pathways of OA or by using stem cells, attempts have been made to prevent the destruction of cartilage or to reduce the pain-triggering inflammatory reaction. The development of OA pharmacotherapies is primarily focused on the protection or even the regeneration of cartilage tissue, led by the assumption that the protection of cartilage structure also influences the clinical symptoms. Contradictorily, an MRI-based study has shown that a loss of cartilage thickness is associated with only a small amount of worsening knee pain, an association mediated in part by worsening synovitis [4]. Possible therapeutic approaches are, therefore, in addition to chondroprotection, the causal treatment of synovitis.

The aim of this narrative review is to summarize the current pharmacotherapeutic treatment options for OA therapy beyond the solely symptomatic attempts. Causal and non-causal, as well as future, therapeutic approaches are discussed below. This review attempts to think outside the box by also mentioning the approaches in the field of gene therapy and epigenetics. OA therapies that are not drug therapies are not addressed in this review. Lifestyle modifications, such as a change in eating habits and dietary weight management, as well as activity, physiotherapy, and mechanical issues also play a large part in successful therapy. A claim to the completeness of all possible therapy alternatives for OA can therefore not be made. The question addressed here is whether OA therapy will remain mainly symptomatic or if the pill against arthrosis already exists?

## 2. Symptomatic OA Therapy

### 2.1. Pain Killers

The most common therapy for OA is still the use of painkillers. In principle, there are different classes of substances available. These drugs are a purely symptomatic therapy that can alleviate but cannot influence the progression of OA. As this topic has been well described in several guidelines and in a recent narrative review [2], just a short summary of the most used painkillers will be presented in the following.

Nonsteroidal anti-inflammatory drugs (NSAIDs) are the most commonly used and prescribed pain medications in the treatment of OA and are routinely recommended in clinical practice guidelines [5]. Pain management is an essential part of OA treatment. Inflammation occurs in all joint tissues and is thus an essential target of drug therapy. Inflammation leads to the release of various neuromodulatory mediators such as cytokines and prostaglandins which are synthesized by the enzyme cyclooxygenase (COX). The release of proinflammatory mediators leads to the classic symptoms of synovitis, joint swelling and hyperthermia. Furthermore, inflammation can induce joint tissue damage [6,7,8].

NSAIDs can counteract these processes by reducing the corresponding inflammatory cascades by inhibiting the COX enzyme. Additionally, NSAIDs can reduce the sensation of pain relief by desensitizing nociceptors and are thus effective drugs in the symptomatic therapy of OA [9]. Smith et al. and Stewart et al. report in their meta-analysis that oral NSAIDs are similarly effective as opioids in relieving pain in patients with OA [10,11]. However, the use of NSAIDS is limited due to their known but rare risk of gastrointestinal, cardiovascular, and renal adverse events [12].

Studies have concluded that the oral use of NSAIDs is mainly recommended for short-term or intermittent therapy, rather than prolonged treatment [13]. Osani et al. reported in their meta-analysis that the therapeutic peak of NSAID-induced benefits in patients with knee OA was reached after 2 weeks and decreased over time, while cardiovascular and gastrointestinal side effects were already significantly increased after 4 weeks of treatment.

The usage of selective COX-2 inhibitors has increased in the past decades, due to their benefit of combining both anti-inflammatory and analgesic properties, while providing better gastrointestinal tolerability and a reduction in gastroduodenal ulcers in comparison with non-selective COX inhibitors [14].

Coxibs, however, have been associated with an increase in cardiovascular events due to a reduction in prostaglandin synthesis [15]. On the contrary, in a meta-analysis conducted by Cooper et al., recent data has shown that celecoxib does not significantly increase cardiovascular risk when compared with conventional NSAIDs and placebos, regardless of the dose and the duration of treatment [16]. However, recommendations on the analgesic use of various NSAIDs for patients with underlying health conditions remain conflicting. When prescribing any type of NSAIDs, all risk factors (age, gastrointestinal, cardiovascular, and renal events) should be taken into account. Due to adverse effects, NSAIDs are not intended for long term treatment. 

A simple and effective way to reduce the risk of gastrointestinal side effects is the topical application of NSAIDs. Amemiya et al. compared the effects of esflurbiprofen patches versus flurbiprofen tablets in patients with gonarthrosis in 2021. Maximum esflurbiprofen concentrations were observed in the synovium, synovial fluids, and plasma after esflurbiprofen plaster (SFPP) application for 12 h. The numeric rating scale (NRS) results indicated a long-lasting effect of SFPP. Through transdermal application, a continuously high drug effect level was achieved. Overall, no dose peaks need to be accepted to achieve the same effect as with repeated daily oral administration [17].

Depending on the risk profile and the patient’s pain perception, it must be decided on a case-by-case basis whether oral or topical application is preferable.

Opioids are an alternative for more severe pain states when usual pain medication or other pain treatments are not sufficient or may not be used. In a direct comparison with NSAIDs, tramadol was inferior in terms of analgesia [18]. When prescribing opioids, the side effect profile, such as central nervous effects with fatigue, dizziness, and impaired balance, should be taken into account. This is where opioids differ adversely from NSAIDs. Opioids such as tramadol should not be prescribed as a first-line therapy. If other drugs are not used due to their side effect profile or if a non-drug therapy (e.g., surgery) is not currently available, they should be used. Opioids are not used long-term or routinely for osteoarthritis. However, they may be indicated for short-term therapy [2].

### 2.2. Symptomatic OA Therapy with Potentially Causal Impact

#### 2.2.1. Hyaluronic Acid (Intra-Articular)

Hyaluronic acid (HA) is reduced in both concentration and molecular size in patients with osteoarthritis; hence, the intra-articular injection of HA aims to substitute the physiological synovial HA. The individual hyaluronic acid (HA) products differ in terms of production, molecular weight, degree of cross-linking, viscosity, and frequency of application per series. Despite a large number of scientific studies, the effectiveness of this therapy is still disputed in the literature. While some meta-analyses show tangible benefits, they often include studies with a high risk of bias. When meta-analyses are restricted to studies with a low risk of bias, the effect of IAHA is similar to that of saline injections [2,19]. There is a growing amount of literature demonstrating that product differences, particularly HA molecular weight, may have a significant effect on treatment outcomes, with a higher molecular weight showing better results [19]. Due to the invasive mode of application, the indication for intra-articular HA injection should, however, only be made when the prescription of NSAIDs is not possible due to side effects or contraindications or when these are not sufficiently effective. The possible side effects such as joint infection or irritation of the knee joint must be discussed with the patient in advance; severe adverse events are rare.

#### 2.2.2. Corticosteroids (Intra-Articular)

The intra-articular injection of a glucocorticoid for the relief of the acute inflammatory symptoms of activated arthrosis can be a useful measure. The aim of the treatment is to reduce pain and restore mobility. Randomised and placebo-controlled studies have shown that the intra-articular injection of a glucocorticoid into an osteoarthritic knee joint can significantly reduce symptoms over a period of at least 1 week. Occasionally, however, a prolonged effect lasting 16–24 weeks is observed after the intra-articular application of glucocorticoids [20]. They should be used in the short term at the lowest possible but effective dosage for painful arthrosis that does not respond to other therapeutic measures. This may be the case, for example, in inflamed arthrosis with acute pain exacerbation.

There is a lack of data on the long-term effects of cortisone injections on articular cartilage and on a possible association with adverse joint effects. However, in some in vivo studies, corticosteroids were found to be cytotoxic to articular cartilage [21]. In a 2-year randomised trial, the cortisone-infiltration group showed higher cartilage loss than the saline-injection group. Taken together, cortisone injections are still a common treatment option, but they are not without side effects. Frequent use must therefore be considered critically [22]. Injections at too short intervals increase the risk of infection. The patient should be informed about the possible complications of infection and/or tissue atrophy as well as about possible treatment alternatives.

#### 2.2.3. Platelet-Rich Plasma (Intra-Articular)

Aside from the available oral drugs against OA, the usage of autologous growth factors, e.g., intra-articular injections of platelet-rich plasma (PRP), can be used for OA treatment, especially for knee OA. The autologous fluid, which is obtained by centrifuging whole blood, is a highly concentrated cocktail of inflammatory mediators and growth factors capable of reducing inflammatory distress and stimulating cell proliferation and cartilaginous matrix production [16]. Between 2011 and 2021, 867 studies on the topic of PRP were published, with an upward trend over the years [23].

Multiple studies have confirmed effective pain relief and the improvement of physical function after PRP injections as well as an acceptable safety profile [24,25].

PRP injections also showed stronger effects compared with conventional injections with corticosteroids and hyaluronic acid [24]. Furthermore, regular injections of intra-articular corticosteroid can lead to the loss of cartilage structure and thus more rapid disease progression.

Consequently, PRP injections may not only contribute to pain relief through anti-inflammatory effects but can also provide lasting pain relief and functional restoration through targeted structural reconstruction when used over an extended period of time.

It was shown that PRP injections significantly improved physical function and WOMAC scores at 3, and up to 12, months [24].

Patients undergoing treatment with PRP injections experience both pain relief and improved joint function. However, it remains unclear whether the short-term effect of PRP injection is due to the temporary changes in the joint environment or whether PRP injections actually lead to structural changes, thus preventing the progression of OA. Another unresolved question surrounds which components of PRP cause this effect. In particular, the proportion of leukocytes (leukocyte-rich or leukocyte-poor) is still the subject of research. The first positive results have been achieved with leukocyte-poor plasma [26]. In summary, the use of PRP is showing very encouraging preliminary results; however, its use is not yet recommended as first-line-therapy in the guidelines.

#### 2.2.4. Chondroitin and Glucosamine

Chondroitin and glucosamine have chondroprotective, analgesic, and anti-inflammatory effects. They are symptomatic, slow-acting drugs used against osteoarthritis. Glucosamine is a component of glycosaminoglycans and can be found in high amounts in articular cartilage and synovial fluid. Chondroitin is found in the extracellular matrix of articular cartilage and plays a role in maintaining osmotic pressure. Thus, it could improve elasticity and the resistance of cartilage [27]. Both chondroitin and glucosamine seem to develop their effects—partly in different forms—through the use of many different pathways. However, only a few selected ones will be mentioned below. Glucosamine was shown to decrease the levels of proinflammatory interleukin-1 (IL-1), interleukin-6 (IL-6), tumor necrosis factor-α (TNF-α), and C-reactive protein (CRP) in studies with rats. In contrast, the anti-inflammatory interleukins IL-2 and IL-10 were increased [28,29,30]. Glucosamine also appears to have immunomodulatory effects that affect the activity of phospholipase A2, matrix metalloproteinases, or aggrecans [31]. Moreover, chondroitin and glucosamine block the pathways involved in inflammation in osteoarthritis, such as the mitogen-activated protein kinase (MAPK) pathway [32]. In addition, both chondroitin and glucosamine have antioxidant effects [33].

In a meta-analysis by Zhu et al. from 2018, it was shown that chondroitin—via oral administration—significantly alleviates pain and leads to an improvement in physical function compared with a placebo. Glucosamine, on the other hand, may improve stiffness [34]. A combination of glucosamine and chondroitin appears to provide better pain relief than acetaminophen in hip and knee OA, but celecoxib showed the best results in this study [35]. It was found that both chondroitin alone and glucosamine alone could significantly reduce the decrease in joint space. Moreover, intraarticular injections of hyaluronic acid in combination with glucosamine hydrochloride led to a significantly higher reduction in IL-6, IL-1β, and TGF-β compared with hyaluronic acid alone in patients with temporomandibular OA [36].

Kwoh et al. and Fransen et al. failed to demonstrate any changes in joint structure with chondroitin or glucosamine administration in patients with chronic knee pain in OA [37,38]. Another meta-analysis summarized seventeen studies, of which only seven studies demonstrated a statistically significant reduction in pain and four studies demonstrated a reduction in joint space narrowing [39].

Several smaller dosages of glucosamine throughout the day appear to be more effective than one large dose per day [40].

Taken together, chondroitin and glucosamine seem to have an effect on the milder forms of OA, reducing joint inflammation and pain. The administration is safe and shows only a small number of adverse effects, such as headache or nausea. Overall, however, there are conflicting results regarding their clinical efficacy. Thus, in patients with contraindications to NSAIDs or with an increased risk of gastrointestinal or cardiovascular risks, the use of oral glucosamine and chondroitin may be considered as a treatment trial before more invasive therapies are undertaken.

#### 2.2.5. Collagen

Collagen is a protein of the extracellular matrix that occurs mainly as collagen type II in the articular cartilage. The enteral absorption of undenatured type II collagen is very low, but di- or tripeptides containing the amino acids proline or hydroxyproline can be absorbed and show an effect [41]. Hydrolyzed collagen could contribute to cartilage regeneration by increasing the synthesis of macromolecules in the extracellular matrix [42]. In addition, collagen is able to modulate both humoral and cellular components of the immune system. It contributes to the body’s ability to distinguish between harmless molecules and potentially harmful pathogens [43]. This leads, for example, to the transformation of naive T cells into T regulatory cells that produce anti-inflammatory substances such as TGF-β and IL-10 [44]. Proline and hydroxyproline can induce hyaluronic acid synthesis [45] and the chondrocytes to synthesize glycosaminoglycans [46].

Pain in patients with hip and knee OA can be alleviated using an oral supplementation of collagen. WOMAC scores, VAS scores, and quality of life improve significantly compared with a placebo [47]. Trc et al. compared a supplementation of hydrolyzed collagen and glucosamine sulphate for 90 days, respectively. The supplementation of hydrolyzed collagen led to a statistically significant improvement in WOMAC and VAS scores compared with glucosamine sulphate [48].

Joint conditions seem to improve following the administration of collagen. It may induce cartilage repair to maintain structure and function. The clinical use of collagen is safe and has minimal adverse effects, mainly gastrointestinal. However, further studies are needed to show the benefits in the treatment of patients with OA and to determine the optimal dosage and duration.

## 3. Causal OA Therapy

### 3.1. Monoclonal Antibodies

Over the last few decades, monoclonal antibodies have emerged as a revolutionary tool in the field of medicine with many promising clinical applications. One of the biggest advantages of monoclonal antibodies is their high specificity as they are designed to target molecules in the body such as cytokines, growth factors, and receptors. In the context of osteoarthritis, antibodies revolutionized the treatment of rheumatoid arthritis. Currently, drug therapy (“the pill against”) has been established to slow down or even stop the progression of the autoimmune disease and has caused the end of the vast majority of formerly regular severe joint destructions. Although the effects and goals of treatment are partly comparable to OA, they are not established as standard therapy. The question arises, will this change? Up to now, there have been some approaches that use monoclonal antibodies in pain therapy of OA, some of which will be presented in the following.

#### 3.1.1. TNF and IL-1 Inhibitors

In one randomized controlled trial, patients with erosive hand OA were treated with the anti-TNF antibody adalimumab (subcutaneous administration once a week) or a placebo for 12 weeks each. Pain intensity was measured with the VAS score. There was no significant difference [49]. Another anti-TNF antibody (etanercept) failed to demonstrate any benefit in a treatment duration of 24 weeks compared with a placebo in hand OA [50]. Canakinumab is an IL-1 inhibitor that showed a reduced rate of joint arthroplasties in patients with atherosclerotic disease [51]. However, further studies failed to show any benefits with respect to pain alleviation in patients with OA when IL-1 was blocked [52]. Overall, TNF and IL-1 inhibitors seem to be rather unsuitable for patients with OA.

#### 3.1.2. Anti-NGF

Joint tissues have been innervated using nociceptors, except for cartilage. Nerve growth factor is an important neurotrophin in inflamed synovium. It is upregulated in patients with OA and leads to an increase in pain. There are three different monoclonal antibodies used in therapy: Tanezumab, Ulranumab, and Fasinumab. They lead to impressive pain relief in patients with knee and hip OA but accelerate the progression of OA [53]. The administration of fewer doses showed a reduced but still substantial effect on pain and function. Nevertheless, 3% of the patients suffered from progressive OA [54]. There are a few other adverse effects such as peripheral neuropathies, headaches, upper respiratory tract infections, oedema, or joint pain [55]. NGF seems to be a relevant factor for cartilage integrity or the repair of cartilage, so that a complete blockade is not an effective treatment in patients with OA. Anti-NGF treatment is promising, but studies are needed to find the optimal dosage to alleviate pain and reduce the adverse effects.

### 3.2. Stem Cell Treatments

Mesenchymal stem cells (MSCs), as a specific type of adult stem cell, possess great potential in regenerative therapy due to their capacity for self-renewal and differentiation [56]. Great attention has been paid to cell-based therapy that may influence cartilage repair such as mesenchymal stem cell therapy. Most studies have been conducted in the context of knee joint osteoarthritis. MSCs are primarily used as intra-articular injection therapy. MSCs modulate immune or inflammatory effects and tissue regeneration in knee osteoarthritis [57,58]. The exact mechanism of MSC therapy remains unclear. It is known that cartilage repair and protection against OA-induced cartilage degeneration is promoted by MSC-derived extracellular vesicles.

Injected MSCs are expected to repair damaged issues due to the trilineage potential and immunomodulatory properties of MSCs. MSCs can be harvested from different sites. The best known or most accessible sites are bone marrow or fat tissue. Other sources include muscle tissue, synovial membranes, or placenta. In addition, the cells can be obtained either from autogenic or allogenic sources. The advantage of allogenic stem cells is that they can be harvested from healthy donors and expanded in vitro to obtain a clinically relevant amount for injection. The disadvantage of allogeneic cell collection is a possible reaction of the recipient’s immune system after injection [59].

Studies in humans have reported variable structural outcomes after MSC injection from hyaline-like cartilage to fibrous tissue. A meta-analysis including 582 knee-OA-patients in 11 trials was performed to assess the efficacy and safety of MSC treatment for knee OA patients using VAS, IKDC, WOMAC, Lequesne, Lysholm, and Tegner scores. MSC-treatment groups from the identified trials were compared with their respective control groups. It shows that VAS decreases and IKDC increases significantly after 24 months follow up. MSC therapy also showed significant decreases in WOMAC and Lequesne scores after the 12-month follow up. The evaluation of Lysholm (24-month) and Tegner (12- and 24-month) scores also demonstrated favorable results for MSC treatment. The effects of MSC therapy on short-term primary endpoints still need to be evaluated in a larger number of patients [60].

Another important question is the dosage at which the stem cells should be injected. A larger amount of injected MSCs may be expected to induce better effects. Interestingly, in studies with allogeneic stem cells, it was found that no improvement was observed in relation to “high dose” as opposed to “low dose” stem cell transplantation. The clinical symptoms and MRI imaging of the cartilage were the main factors assessed. There were also differences in the dose effect of stem cells depending on their origin. These results suggest that appropriate MSC doses applied in intra-articular injections to OA patients need to be determined for each origin of MSCs [61,62]. Furthermore, MSC injection combined with other agents such as hyaluronic acid [63] or PRP [64] has better therapeutic effects than MSC injection alone. This implies the possible value of drug cocktail therapy when using MSC injection in knee OA patients.

Overall, MSC transplantation treatment was shown to be safe and has great potential as an efficacious clinical therapy, especially for patients with knee OA. Further clinical and in vitro studies are needed to better clarify the underlying molecular and biochemical mechanisms. Particularly, it is yet to be determined whether MSCs should be injected as a single agent or in combination with another drug or as a complementary therapy to surgical treatment.

## 4. Future Directions

### 4.1. Gene Therapy

Gene therapy consists of using a vector to bring genes directly into cells and tissues to treat a specific disease. Viral vectors include RNA viruses and DNA viruses. Two different gene therapy strategies are currently in preclinical and clinical development for OA [65]. The first approach consists of ex vivo modifying and amplifying cells, followed by their intra-articular injection. The aim is to over-express TGF-ß-1 in irradiated allogenic chondrocytes [66]. The second approach is an in vivo gene therapy through the local or systemic injection of viral vectors containing the transgene of interest. In general, OA gene therapy aims to reduce inflammation through overexpressing transgenes such as IL-1Ra or a soluble TNF receptor [67]. In the future, gene therapy could become a strategy to regulate the intra-articular expression of therapeutic targets in OA.

### 4.2. Epigenetics

Epigenetics is a field of research that analyzes the changes in gene expression or cell phenotype occurring without the modification of the DNA sequence. Several epigenetic regulators appear to be involved in the pathogenesis of OA. The epigenetic profiling of articular chondrocytes has revealed the existence of an activating sequence that is present in billions of people with a risk locus (GDF5-UQCC1) that is involved in OA progression. These epigenetic modifications can also suppress the expression of protective genes in OA [61]. Abnormal changes in DNA methylation occur in the promoter regions of related genes and signaling pathways in OA chondrocytes. Epigenetic regulation typically involves DNA methylation, histone modification, and noncoding RNA-mediated regulation. Epigenetic mechanisms can control several signaling pathways simultaneously. For this reason, epigenetic modifications have been considered a potential therapeutic target to manage OA [68].

## 5. Conclusions

OA is caused by an imbalance between degenerative and regenerative processes. To date, therapy is still mainly symptomatic. As well as improving mechanical issues, the future therapeutic goal must be to gain an even better understanding of the cellular and molecular causes of OA. Non-surgical therapy comprises basic measures such as weight reduction, exercise therapy (water and land), and health education. Specific measures comprise biomechanical interventions, physiotherapy, physical measures, and drug therapy. Several pharmacologic, genetic, and even epigenetic attempts are promising, but unfortunately, so far none have proven causal therapy to work or cure OA. The early detection of osteoarthritis by means of biomarkers and imaging must also gain focus to allow for early and targeted treatment.

With regards to drug therapy, the individual risk profile as well as the level of suffering or pain intensity must be taken into account before treatment is started. For this reason, it is not possible to give general advice on drug therapy for OA. A non-causal but proven treatment option for OA is the use of painkillers (mainly NSAIDs), which are beyond the focus of this review. Their duration of therapy is limited due to side effects, especially in patients with corresponding underlying diseases. They are solely symptomatic, therefore alleviate but do not alter the progression of OA. The use of PRP injections seems to clearly overcome hyaluronic acid which has recently shown conflicting results. PRP can potentially stimulate cell proliferation and cartilaginous matrix production and provide lasting pain relief and functional restoration through targeted structural reconstruction when used over an extended period of time. Therapies with chondroprotective substances such as chondroitin, glucosamine, collagen, or monoclonal antibodies lead to a reduction in pain. However, a significant therapeutic effect in singular application has not been detected so far. The use of stem cells in arthrosis therapy, however, is a promising therapy. Its possibility for cell regeneration or conversion into functional cells holds great potential, especially in the context of the therapy of degenerative diseases such as OA. Favored cell sources, dosage, and therapy duration remain unclear.

Due to the multifactorial genesis of OA, most therapeutic approaches are still symptomatic and the “causal pill against” OA does not yet exist. Future therapeutic approaches have to identify innovative therapeutic targets aimed at influencing the inflammatory and metabolic processes underlying the pathogenesis and progression of OA.

## Data Availability

Not applicable.

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
