# Peer review of "Pharmatherapeutic Treatment of Osteoarthrosis—Does the Pill against Already Exist? A Narrative Review"

_jpm, 2023, doi:10.3390/jpm13071087_

Round 1

Reviewer 1 Report (New Reviewer)

Comments to Author:

The review article covers contemporary knowledge on treatment for osteoarthritis.

In the main document, there are several points needing revision.  

1.       On page 3, line 47: A period for the first sentence is missing.

2.       On page 4, line 6: duplication ‘are are’

3.       The potential adverse effect of joint infection, discussed in 2.2.1. hyaluronic acid, should be also discussed in 2.2.2. intra-articular corticosteroids because corticosteroids increase  risk of infection.

4.       On page 8, line 16 and line 23: ‘osteoarthritis’ should be abbreviated to ‘OA’.

Author Response

Thank you for the helpful comments. Below you will find the changes listed.
Both the linguistic and the substantive changes are deposited in the paper.

  1. On page 3, line 47: A period for the first sentence is missing.

The punctuation mark was added.

  1. On page 4, line 6: duplication ‘are are’ 

There is an “r” missing. The sentence should read “are rare”.

  1. The potential adverse effect of joint infection, discussed in 2.2.1. hyaluronic acid, should be also discussed in 2.2.2. intra-articular corticosteroids because corticosteroids increase  risk of infection.

Thank you for the comment. As shown below, a paragraph on the risk of infection has been added.

L 170-173: “Injections at too short intervals increase the risk of infection. The patient should be informed about the possible complications of infection and/or tissue atrophy as well as about possible treatment alternatives.”

  1. On page 8, line 16 and line 23: ‘osteoarthritis’ should be abbreviated to ‘OA’.

Osteoarthritis has been changed to OA

Reviewer 2 Report (New Reviewer)

the article conclude the drug used nowadays to treat OA, reviewed and narrate contemporary, and list some future treatment trends.

minor issues:

1. could add a conclusive sentence after very kind of drugs, and transitional paragraph needed to connect them in a logic or chronological way.

2. some discussion should be made to talk different methods pros and cons, better illustrated them as tables or figures outputs.

please polish up the English expressions.

Author Response

Thank you for the helpful comments. Below you will find the changes listed.

Both the linguistic and the substantive changes are deposited in the paper.

minor issues:

  1. could add a conclusive sentence after very kind of drugs, and transitional paragraph needed to connect them in a logic or chronological way.

Thank you for the comment which we highly appreciate. We discussed this with other reviewers which however led us to focus on treatment options beyond solely symptomatic attempts and beyond descriptions which can be found in well established (inter-) national guidelines.

Please therefore find the last paragraph in the Introduction which summarizes this:

“The aim of this narrative review is to summarize the current pharmacotherapeutic treatment options for OA therapy beyond solely symptomatic attempts. Causal and non-causal as well as future therapeutic approaches are discussed below. The review tries to think outside the box by also mentioning the approaches in the field of gene therapy and epigenetics. OA therapies apart from drug therapy were not addressed in the review. Of course, a change in eating habits, dietary weight management as well as activity, physiotherapy and mechanical issues also play a large part in successful therapy. A claim to the completeness of all possible therapy alternatives for OA can therefore not be raised of course. The question was, whether therapy still stays mainly symptomatic or if the pill against arthrosis already exists?”

In order to make the structure logical for the reader, the points have not been ordered "chronologically" but according to therapy approach.

here the structure was made into the sub-items:

  1. symptomatic OA therapy
    2.2 Symptomatic OA therapy with potentially causal impact
    3. causal OA therapy

For ease of reference, the words "intra-articular" have been added to each of the following chapters.

2.2.1. hyaluronic acid (intra-articular)
2.2.2. corticosteroids (intra-articular)
2.2.3. platelet rich plasma (intra-articular)

  1. some discussion should be made to talk different methods pros and cons, better illustrated them as tables or figures outputs.

Thank you for the comment. An overview of the therapeutic options (divided into injection and oral administration) can be found in the "Graphical Abstract" which accompanies the paper. Here, possible disadvantages are also listed briefly.

In our opinion, no general therapy advice can be given, as the patient's risk profile and the degree of suffering must be discussed individually in each case. We edited the conclusion accordingly:

Line 382-385: “Non-surgical therapy comprises basic measures such as weight reduction, exercise therapy (water and land) and health education. Specific measures comprise biomechanical interventions, physiotherapy, physical measures and drug therapy.“

And line 389-391: “With regard to drug therapy, the individual risk profile as well as the level of suffering or pain intensity must be taken into account before treatment is started. For this reason, it is not possible to give general advice on drug therapy for OA.”

This manuscript is a resubmission of an earlier submission. The following is a list of the peer review reports and author responses from that submission.

Round 1

Reviewer 1 Report

This is a narrative review of some of the treatments available for the management of osteoarthritis.

The introduction is well constructed, and the alternatives described are, in general, well supported bibliographically. However, some of the currently available therapeutic options are missing, without the reader knowing the reason for these omissions. Perhaps the reason exists, but this reviewer has not found it. It is only necessary to take a look at the guidelines and recommendations issued by the scientific societies to realize that there are treatments that are not addressed in this manuscript, such as corticosteroids, hyaluronic acid, duloxetine, gabapentin, weak opioids........ while other treatments that are not recommended by the guidelines, such as PRP, are addressed in the manuscript. In my opinion, there are 2 possible options: to argue clearly the reasons for these omissions or to include the missing treatments.

On the other hand, all of us involved in clinical research know, that in this field, there is always more that can be done and aspects to be considered, but this does not justify authors to end almost all narratives in the same way: "further studies are needed............" in many cases certainties are available that allow to issue a recommendation. They should end in a different way, if necessary by drawing on existing recommendations or on the working protocols of their hospital.

English is in general fine.

Please change "Eterocoxib" to "etoricoxib" (line 100)

Reviewer 2 Report

The authors summarized the current pharmacotherapeutic treatment options for OA therapy. There have been many reviews on OA therapy, and in order to make this review more novel, it is hoped that the authors can discuss about some novel or emerging therapeutic drugs.

The following issues should be addressed. 

Major:

1.     Need to add a part about patches for external use on OA, like Flurbiprofen Cataplasms, etc.

2.     Illustrate a diagram outlining all strategies for treating OA.

3.     Section 3. Chondroitin and Glucosamine and Section 4. Collagen should be placed in Section 2.2. Symptomatic OA therapy with potentially causal impact.

4.     Discuss about other treatment options for future directions, such as cartilage regeneration or other matrix related technologies. 

Minor:

1.     Grammar: line 97, “In”; line 118, “Symtomatic”; line 155, “Ih”.

2.     Font: line 122-125, “The autologous fluid, which is obtained by centrifuging whole blood, is a highly concentrated cocktail of inflammatory mediators and growth factors capable of reducing inflammatory distress and stimulating cell proliferation and cartilaginous matrix production [17].”; line 150, “cartilaginous matrix production”.

3.     Punctuation mark: line 210, dot after “gastrointestinal”; line 225, spacing in “[45] .”.

There are some grammar, font and punctuation mark errors that need to be carefully corrected.

Reviewer 3 Report

Journal

Journal of personalized medicine

Title

 The pill against arthrosis? A narrative review

OA is a widespread disease that affects a large number of populations. We are looking forward for any new development of specific drugs for this disorder.

The idea of the manuscript is interesting. However, the writing is to some extent not scientific, and the manuscript needs to be modified.

Comments

The title mentioned “the pila”. However the authors include invasive interventions .What was meant by the word “pill”

12.. for osteoarthritis…. appropriate use of abbreviations is required.

17.. medicine and and in…remove one “and”

17.. style of treating rheuma-17 toid arthritis. The authors meant osteoarthritis.

The research gap should be clear and concise.

102.. Coxibs, however, have been associated with an increase of cardiovascular events due 102 to reduction in prostaglandin synthesis [16] …some paragraphs are too short which nit cope with the scientific writing.

109.. NSAIDs can provide sufficient pain relief in patients with OA, and selective COX-2 109 inhibitors specifically, have…each paragraph should include a different idea, not a repetition of what have been mentioned in the previous paragraphs.

118.. PRP (Platelet rich plasma) ...the authors start with NSAIDs then proceed to interventional procedure without mentioning other noninvasive pharmacological therapy.

118.. The results of PRP are still conflicting and PRP is still not recommended as an effective modality in the treatment of OA.

Again, the writing manner needs more editing to be in a scientific manner.

Again, the arrangement of titles and subtitles is not appropriate.  

Moderate editing of English language

Round 2

Reviewer 1 Report

Although there are several reviews on the therapeutic options available for the treatment of osteoarthritis, it is always interesting to have new contributions on this topic. I appreciate the authors' efforts to improve the manuscript, but I regret to say that the new version does not reach the expected results.

Some minor mistakes present. For instance typical instead of topical. But this is not a key issue.

Reviewer 2 Report

I am still not satisfied by the current version of the manuscript. The novelty of the manuscript is still insufficient, and some chapters have not been elaborated on. The chapters in 2.1.1 NSAIDs and 2.2.3 PRP are not well organized. The diagram outlining all the strategies treating for OA is still missing. 

Some formatting issues, such as 2.1. paint killers, 2.1.2 opioids, etc., where the initial letters are not capitalized.

Reviewer 3 Report

The writing of the manuscript does not sound scientific.The authors did not address all the reviewer comments 

Moderate editing of English language